# Correlation Comparison and Personalized Utility of Field Walking Tests in Assessing the Exercise Capacity of Patients with Chronic Obstructive Pulmonary Disease: A Randomized Controlled Trial

**DOI:** 10.3390/jpm12060901

**Published:** 2022-05-30

**Authors:** Eun Jae Ko, Jang Ho Lee, Hyang Yi Lee, Seong Ho Lee, Hack-Jae Lee, Ganghee Chae, Sei Won Lee, Seung Won Ra

**Affiliations:** 1Department of Rehabilitation Medicine, Asan Medical Center, University of Ulsan College of Medicine, Seoul 05505, Korea; ejko.amc@gmail.com; 2Department of Pulmonology and Critical Care Medicine, Asan Medical Center, University of Ulsan College of Medicine, Seoul 05505, Korea; leejh1342@naver.com; 3Department of Pulmonary Physical Therapy, Asan Medical Center, Seoul 05505, Korea; lhy4981@hanmail.net (H.Y.L.); dltjdgh1234@naver.com (S.H.L.); 4Department of Pulmonary Rehabilitation, Ulsan University Hospital, Ulsan 44033, Korea; leehj@uuh.ulsan.kr; 5Division of Pulmonology and Critical Care Medicine, Department of Internal Medicine, Ulsan University Hospital, University of Ulsan College of Medicine, Ulsan 44033, Korea; margiela07@naver.com

**Keywords:** 6 min walking test, chronic obstructive pulmonary disease, incremental shuttle walking test, validity

## Abstract

Background: Incremental shuttle walking tests (ISWT) are regarded as valuable alternatives to 6-min walking tests (6MWT) and cardiopulmonary exercise tests (CPET) owing to the maximal and externally paced loading. This study investigated the validity and reliability of ISWT by analyzing the correlation of the distances of two field tests with peak oxygen consumption (VO_2_) of CPET in patients with COPD. Methods: In this randomized controlled trial, patients with COPD were enrolled from two hospitals. Three assessments were performed for all patients. The ISWT and 6MWT were repeated twice in Hospital 1 to assess reliability. Results: A total of 29 patients were enrolled. The distances of ISWT (0.782, *p* < 0.001) and 6MWT (0.512, *p* = 0.005) correlated with peak VO_2_. The intraclass correlation coefficients of both ISWT (0.988, *p* < 0.001) and 6MWT (0.959, *p* < 0.001) was high. Patients with higher peak VO_2_ walked a longer distance in ISWT than 6MWT (*r* = 0.590, *p* < 0.001). Conclusions: The ISWT more highly correlates with peak VO_2_ than the 6MWT and has excellent reliability in patients with COPD. According to peak VO_2_, the walking distances of each field test varied, suggesting that the application should be personalized for the exercise capacity.

## 1. Introduction

Exercise capacity influences the survival of patients with chronic obstructive pulmonary disease (COPD) [1]; thus, its improvement is essential in COPD management [2]. Therefore, assessing exercise capacity beyond pulmonary function is vital to predict prognosis and evaluate the effects of pharmacologic and nonpharmacologic treatment, including inhaler therapy, exercise training, and pulmonary rehabilitation [3].

In clinical settings, the cardiopulmonary exercise test (CPET) and field walk tests, such as the 6 min walking test (6MWT) and shuttle walking test, are typically used to assess the exercise capacity in patients with chronic respiratory diseases. Despite being the gold standard [4], CPET is not widely available and requires a gas analyzer and trained personnel, and often patients with cardiorespiratory diseases find it hard to exercise with a mouthpiece. The 6MWT is a self-paced test susceptible to a substantial training effect and is sensitive to alterations in methodology, such as encouragement or oxygen supplementation [3], while in turn the protocols are difficult to standardize and may be overly influenced by motivation. Moreover, 6MWT needs a 30 m corridor [5] and is difficult to implement in relatively small institutions. In contrast, the incremental shuttle walking test (ISWT) is an externally paced and incremental test using a simple and easy-to-use 10 m course [3,6], highly correlates with the peak oxygen consumption (VO_2_), and is related to the hospitalization and survival of many respiratory or cardiac patients [7,8]. Singh et al. [6] first introduced ISWT in 1991 and then reported its validity, reliability, and minimal clinically significant difference [7].

Although the ISWT has multiple strengths, it has not been broadly introduced in clinical settings. Moreover, limited research has been conducted on the validity of ISWT, especially in comparison to ISWT and 6MWT with CPET in COPD patients. Hence, this study assesses the ISWT validity and reliability by analyzing the correlation of distances of 6MWT and ISWT with the peak VO_2_ of the CPET in patients with COPD. 

## 2. Materials and Methods

### 2.1. Study Design

This was a randomized controlled trial—a multicenter prospective correlation study. After clinical assessment, patients were randomly assigned to the six test order groups using randomly generated treatment allocations with sealed opaque envelopes created by an impartial researcher. Owing to the nature of the intervention, patients were aware of their test orders. According to the group assigned, three assessments (CPET, 6MWT, and ISWT) were performed at 2-day intervals on all patients (Figure 1). The Institutional Review Board (IRB) of Ulsan University Hospital (UUH; IRB file no.: 2019-11-010) and Asan Medical Center (AMC; IRB file no.: 2019-1520) approved this study. The trial has been registered at clinicaltrials.gov (NCT04178278). All patients were informed about the aim and procedure of the study and written informed consent was obtained from all patients before the study. Figure 1 illustrates the study design and flowchart. This study was conducted in accordance with the amended Declaration of Helsinki.

### 2.2. Study Participants

Patients with COPD were enrolled between May 2020 and December 2020 from the Departments of Pulmonology at UUH (Hospital 1) and AMC (Hospital 2). The inclusion criteria were as follows: (i) patients diagnosed with having COPD; (ii) aged >40 years; and (iii) patients with documented airflow limitation, defined as post-bronchodilator forced expiratory volume in 1 second (FEV_1_)/forced vital capacity (FVC) < 0.7. The Appendix A details the exclusion criteria.

### 2.3. Cardiopulmonary Exercise Test

The CPET was performed on a cycle ergometer using the modified Bruce protocol. Throughout the test, 12-lead electrocardiography was recorded continuously, and patients were required to wear a mask and breathe through a calibrated volume sensor. The equipment was recalibrated before each test using a gas calibration mixture of known concentrations. The CPET was conducted such that patients could terminate the test at their discretion if they found it challenging to proceed. The oxygen consumption at the estimated anaerobic threshold (AT) was obtained using the modified V-slope [9] and ventilatory [10] methods. In addition, we evaluated the respiratory quotient (RQ) to assess whether patients had performed sufficiently intense exercise. The test results included peak VO_2_, AT, peak RQ, peak heart rate (HR), peak systolic blood pressure (SBP), peak diastolic blood pressure (DBP), O_2_ pulse, breathing reserve, lowest oxygen saturation (SpO_2_), and maximal voluntary ventilation (MVV). The ratings of perceived exertion (RPE) were expressed on a modified Borg’s scale from 0 to 10.

### 2.4. Six-Minute Walk Test

A single physiotherapist conducted the 6MWT in a 30 m long corridor in each hospital per the standard protocol [5]. After the physiotherapists’ detailed explanation, demonstration, and patients’ brief repetition, two tests were conducted at a 30 min interval in Hospital 1, but only one test was performed in Hospital 2. Patients were instructed to “walk from this end of the corridor to the other at your own pace.” Each minute, the physiotherapist encouraged patients with standardized statements. Patients were permitted to stop and rest during the test but resumed walking as soon as possible. Upon the test completion, the distance walked was determined, and the peak SBP, peak DBP, peak HR, and lowest SpO_2_ were recorded. The RPE was measured on the modified Borg’s scale. The results of the first test were used to analyze the 6MWT validity in Hospital 1.

### 2.5. Incremental Shuttle Walking Test

A single physiotherapist performed the ISWT at each hospital based on the original protocol proposed by Singh et al. [6], and detailed translated protocols were followed at both test sites. After the physiotherapists’ explanation, demonstration, and patients’ practice, two tests were conducted at a 30 min interval in Hospital 1, but only one test was recorded after a practice run in Hospital 2. Patients were asked to walk along a level 10 m course at a previously determined speed dictated by signals; these signals were transmitted to wireless Bluetooth earphones using a smartphone mp3 player (Hospital 1) or an audio speaker using a CD player (Hospital 2). The walking speed progressively increased at 1 min intervals, for a total of 12 levels. The test was terminated if the patient failed to complete the shuttle course in the allotted time. Upon the test completion, the distance walked was measured, along with the peak HR, peak SBP, peak DBP, lowest SpO_2_, and RPE by the modified Borg’s scale. Furthermore, we measured the recovery times of HR, SpO_2_, and Borg’s scale to their baselines. Of note, the results of the first test were used in Hospital 1 to analyze the ISWT validity.

### 2.6. Sample Size Calculation

The power analysis and sample size were calculated using the PASS software version 2019 (NCSS, LLC, Kaysville, UT, USA). We selected a correlation coefficient (*r*) with peak VO_2_ as the primary outcome to validate ISWT based on previous studies [11,12]. There were large variations of the correlation coefficients according to study subjects between ISWT and peak VO_2_ in patients with COPD. Since a new method of transmitting the audio signal for ISWT, i.e., wireless Bluetooth earphones using a smartphone mp3 player in Hospital 1 (vs. an audio speaker using a CD player in Hospital 2) was investigated in this study, we set the correlation coefficient of 0.6 in the most conservative way. A sample size of 24 attains 90% power to detect a difference between the null hypothesis correlation of 0 and the alternative hypothesis correlation of 0.6 using a two-sided hypothesis test with a significance level of 0.05 (*α* = 0.050, *β* = 0.100, *r* = 0.6). We assumed a dropout rate of 20% and planned on enrolling 30 patients with COPD (UUH, 15 patients; AMC, 15 patients) in this study.

### 2.7. Statistical Analysis

All continuous data are presented as the median and interquartile range (IQR) or mean and standard deviation (SD), whereas categorical variables are presented as frequencies and percentages. Categorical variables were analyzed using the *χ*^2^ test or Fisher’s exact test. We used the Kolmogorov–Smirnov test to investigate the normal distribution of continuous variables. We used the Mann–Whitney *U* test for non-normal distributed variables and the Student’s *t* test for normal distributed variables. In addition, we used the Pearson’s correlation for all correlation analyses and compared the correlation coefficients using the Fisher’s *z* transformation between independent groups and Dunn and Clark’s *z* test between the dependent group. Then, reliability analyses were performed using the intraclass correlation coefficient (ICC). In this study, all significance tests were two-sided, and we considered *p* < 0.05 as statistically significant. All analyses were performed using the SPSS software version 24.0 (IBM Corp., Armonk, NY, USA). Statistical comparison of correlations was performed using the R Statistics software version 4.0.2 (The R Foundation, Vienna, Austria) using the “cocor” package [13].

## 3. Results

### 3.1. Participants

A total of 17 patients were enrolled from Hospital 1 (UUH) and 17 patients from Hospital 2 (AMC). Of these, five patients were excluded from the final analysis because they had difficulty walking (*n* = 4) or could not visit the clinic as scheduled (*n* = 1). Finally, 29 patients were enrolled (median age, 67 [IQR, 61–72] years; 28 (96.6%) male). Most of the participants were current or ex-smokers (28/29, 96.6%), with a median pack-year of 35. Hypertension (7/29, 24.1%) was the leading comorbidity. In addition, seven (24.1%) had experienced acute exacerbation in the previous year. Predominant inhalers used were long-acting β_2_ agonist (LABA)/long-acting muscarinic antagonist (LAMA; 17/29, 58.6%), followed by LAMA/LABA/inhaled corticosteroid (ICS; 8/29, 27.6%). The median COPD assessment test (CAT) score was 7.0 (IQR, 4.0–12.0) and the modified Medical Research Council (mMRC) score was 1.0 (IQR, 1.0–2.0). The median FEV_1_ was 1.9 L, with a predicted value of 66.5%, and the median FVC was 3.8 L, with a predicted value of 108.8%. Among 29 patients, the median value of peak VO_2_ was 17.8 mL/kg/min. The mean distances of ISWT and 6MWT were 483.5 m and 525.3 m, respectively. Furthermore, the characteristics were similar between both hospitals, except for inhaler usage and the distances of ISWT and 6MWT (Table 1).

### 3.2. Correlation of CPET with ISWT and 6MWT

Figure 2, Table 2, and Appendix A present the correlation analyses and coefficients. Among all study participants, the distances of ISWT and 6MWT significantly correlated with peak VO_2_, and the correlation coefficient with peak VO_2_ in ISWT was significantly higher than that of 6MWT (ISWT, 0.782; 6MWT, 0.512; *p* = 0.043). In Hospital 1, the correlation coefficients with peak VO_2_ in ISWT was significantly higher than that of 6MWT (ISWT, 0.868; 6MWT, 0.685; *p* = 0.029). Meanwhile, the correlation coefficients with peak VO_2_ in ISWT and 6MWT were similar in Hospital 2 (ISWT, 0.540; 6MWT, 0.572; *p* = 0.883). Because breathing reserve was higher in Hospital 2 than Hospital 1, we reanalyzed the correlation with the exclusion of one patient in Hospital 1 and three patients in Hospital 2 who presented 40% or more breathing reserve (Appendix A). The result of the reanalysis was in line with Table 2. In this analysis, ISWT still presented a numerically higher correlation coefficient than 6MWT, although there was no significant difference (ISWT, 0.783 vs. 6MWT, 0.620; *p* = 0.208).

### 3.3. Reliability of ISWT and 6MWT

Both ISWT and 6MWT were performed two times in Hospital 1, and their reliability was evaluated by ICC. The reliability was very high, and the ICC of the ISWT was numerically higher than that of the 6MWT, with 0.988 (95% confidence interval (CI), 0.965–0.996; *p* < 0.001) and 0.959 (95% CI, 0.871–0.987; *p* < 0.001), respectively (Figure 3). The ranges of the walking distances were wider in ISWT (910–370 m) than 6MWT (645–408 m).

### 3.4. Correlation Analysis of ISWT and 6MWT

Appendix A shows the correlation analyses between ISWT and 6MWT. The mean ratio of the walking distance on 6MWT and ISWT was 1.110 (0.686–1.540; Appendix A). The difference between 6MWT and ISWT negatively correlated with the mean distance in both walking tests (*r* = −0.60, *p* < 0.001; Appendix A). Figure 4 shows the analysis of the correlation between peak VO_2_ and the difference between ISWT and 6MWT. Patients with higher peak VO_2_ walked a longer distance in ISWT than 6MWT (*r* = 0.590, *p* < 0.001).

### 3.5. Clinical Parameters of Each Exercise Test

Appendix A presents the clinical parameters during the exercise tests. Peak SBP, peak DBP, and peak HR were the highest during the CPET (median [IQR]; peak SBP, 193.0 [164.5–216.5], *p* < 0.001; peak DBP, 95.0 [83.5–106.0], *p* = 0.004; peak HR, 130.0 [119.5–142.5], *p* = 0.001) and were consistent in both hospitals. Moreover, dyspnea after exercise tests presented as a Borg scale was the severest during the CPET (median [IQR]; 10.0 [7.0–10.0], *p* < 0.001), also consistent in both hospitals (Appendix A). The lowest SpO_2_ exhibited no significant difference in all three tests.

## 4. Discussion

This multicenter randomized controlled trial demonstrated that the ISWT was more representative of the CPET, the gold standard of exercise capacity, than the 6MWT. In addition, reliability, indicated by the difference between the first and the second test, was better in the ISWT. Importantly, the ceiling effect of the walking test observed in the 6MWT was relatively smaller in the ISWT. The findings of this study are valuable in that all three representative tests to evaluate the exercise capacity were conducted in each patient, illustrating the superiority of the ISWT to substitute the CPET. 

Some previous studies focused on the correlation of ISWT [14,15,16,17,18] or 6MWT [14,17,18,19,20,21,22,23,24,25] with the CPET, but most analyzed the two tests separately and did not conduct all tests together. A study analyzing the three tests together revealed a similar correlation between ISWT and CPET (*r* = 0.73; *p* < 0.001) and between 6MWT and CPET (*r* = 0.73; *p* < 0.001) [14], which somewhat differed from our study. The difference is attributable to the higher FEV_1_ of our participants compared with those of the previous study (mean FEV_1_ percent predicted value: 66.5% vs. 28.9%). The ceiling effect and heterogeneity of the 6MWT is considered more dominant in milder patients [11]. 

In this study, the reliabilities of both ISWT and 6MWT was high (ICC = 0.988, *p* < 0.001 and ICC = 0.959, *p* < 0.001), and both were considered as a reliable measure with a strong correlation between the test and retest results. Previously, the reliability of the ISWT has been mostly examined in patients with COPD. The ICC of the ISWT was reported as 0.88 (95% CI, 0.83–0.92) [26] and 0.89 [27], and the impact of participants’ characteristics, such as age, sex, and disease severity, was relatively small. In addition, the reliability of the 6MWT has been explored in several chronic respiratory diseases. The ICCs ranged in COPD from 0.88 to 0.99 [28,29,30,31], in cystic fibrosis from 0.93 to 0.94 [32,33], and in interstitial lung disease from 0.72 to 0.95 [31,34,35], indicating that diagnostic groups did not exhibit a discernible difference [7]. Moreover, the coefficient variations were very small (0.0475–0.073) in COPD. In our study, the ICCs of both tests were relatively high even compared with the previous studies, suggesting a reliable research environment. Although the ICC of the ISWT was numerically higher than that of the 6MWT, the difference was hard to interpret because of significantly reliable results in both tests.

The CPET requires maximal exercise performance and physical activity using the value from the maximal level of oxidative metabolism [36]. Likewise, the ISWT is considered a maximal exercise test defined by the attainment of HR max ≥90% and peak respiratory exchange ratio ≥1:10 [37,38]. Meanwhile, the 6MWT is historically considered a submaximal test [5]. In a previous study, the 6MWT was inadequate to measure the peak work rate in patients with COPD [39]. In this study, the CPET exhibited the highest peak SBP, peak DBP, peak HR, post-Borg score, and the lowest SpO_2_, consistently followed by the ISWT. These findings suggest that the ISWT elicited higher-intensity exercise than the 6MWT: this intensity and the characteristics during the tests could be related to the difference in the correlation of peak VO_2_ in CPET between ISWT and 6MWT. 

The exercise capacity is a vital prognostic factor of patients with chronic respiratory disease, and precise assessment of the exercise capacity is imperative for clinicians and patients [40]. The maximal oxygen uptake during the CPET is the key parameter for assessing exercise capacity [36]. Nevertheless, inconvenience and the high cost of the CPET hinder broad clinical application as initial and follow-up assessment tools [41]. Although the 6MWT has been examined to substitute the CPET, it has several limitations. For example, the 6MWT has a ceiling effect in mild patients: Vonbank et al. demonstrated that the correlation between CPET and 6MWT was lower in patients with mild COPD [42]. Similar results were reported in chronic heart disease [43]. Noteworthy is that the heterogeneity of ISWT was very low both in mild and severe groups of COPD, whereas a high heterogeneity of 6MWT was found, particularly in mild groups with higher peak VO_2_ [11]. In this study, the ISWT presented a more favorable correlation than the 6MWT, perhaps, because of more maximal exercise performance during the test. Moreover, the ISWT can be performed in a relatively shorter 10 m corridor, indicating it to be a better alternative to the CPET. The BODE index, which is used to predict prognosis in COPD patients, includes 6MWT distance as a variable [1]. Because ISWT presented higher correlation with CPET, further study would be needed to compare ISWT and 6MWT to predict the prognosis of COPD patients, especially in mild COPD patients.

Some differences were presented in the baseline characteristics between Hospital 1 and Hospital 2. Patients in Hospital 1 displayed a longer mean distance of the ISWT than patients in Hospital 2 (538.7 m vs. 424.3 m), but they exhibited a shorter mean distance of the 6MWT (502.5 m vs. 549.8 m), which was quite unexpected. To elucidate this result, we performed further correlation analysis (Figure 4), establishing that patients with higher peak VO_2_ walked a longer distance in ISWT than 6MWT. Although the mean severity of COPD was similar between both study groups (FEV_1_ 66.5% vs. 67.9%), patients with a different severity and heterogenous exercise capacity could have participated in the study. We assume that more patients in Hospital 1 might have had better peak VO_2_ than those in Hospital 2, which led to a better outcome in the ISWT in Hospital 1. Indeed, the patients with higher peak VO_2_ (≥20 mL/kg/min) enrolled more in Hospital 1 (six patients (40.0%) vs. three patients in Hospital 2 (21.4%)). This could have resulted in a higher correlation coefficient with peak VO_2_ in the ISWT in Hospital 1 than in Hospital 2 (*r* = 0.868 vs. *r* = 0.540; Appendix A). Moreover, Hospital 1 used external paced signals using wireless Bluetooth earphones using a smartphone and Hospital 2 used external paced signals using an audio speaker using a CD player in ISWT. Of note, this study was the first attempt to use wireless earphones in the ISWT, showing it was feasible and reliable. Although we could not elucidate the impact of wireless earphones in this study, the utilization of new electronic devices would be more generalized in medical examination, increasing the importance of further study about this issue.

This study has some limitations worth acknowledging. First, a small number of participants could be the main limitation. Nevertheless, we calculated the sample size before starting the study to meet the significant correlation coefficient between ISWT and peak VO_2_ (ClinicalTrials.gov Identifier: NCT04178278). Second, the small sample size also affected an imbalance of baseline characteristics in our study, especially gender. Difference of correlation analysis between the hospitals might have resulted from the difference of baseline characteristics in each hospital. To clarify this point, further investigation with stratified randomization according to gender would supplement our investigation. Third, most enrolled patients had a relatively mild degree of COPD. In recent meta-analysis, a lower correlation coefficient between the distance of field tests and peak VO_2_ and a higher heterogeneity of 6MWT were presented in the mild group than in the severe group [11]. Thus, the ISWT can be considered a more useful alternative to the CPET for assessing the exercise capacity in COPD, especially in the patients with higher peak VO_2_. Further extensive studies, including various severities of COPD and other respiratory diseases, would generalize our results.

## 5. Conclusions

In conclusion, the ISWT has a higher correlation to the CPET than the 6MWT in patients with COPD. According to peak VO_2_, the walking distances of each field test varied, suggesting that the application should be personalized for the exercise capacity. The gap between a self-paced submaximal 6MWT and a more comprehensive CPET could be filled owing to the maximal and externally paced loading of the ISWT. Further study would be needed to compare ISWT and 6MWT as the prognosis factor in COPD patients.

## Figures and Tables

**Figure 1 jpm-12-00901-f001:**
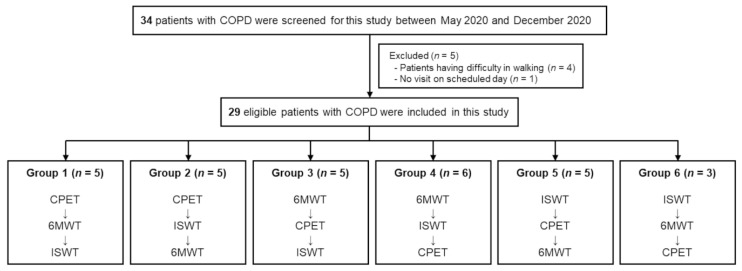
Study flowchart. Three assessments (CPET, 6MWT, and ISWT) were performed in 2-day intervals on all patients, according to the group the patients were assigned. Patients were randomly assigned to the six test order groups using randomly generated treatment allocations with sealed opaque envelopes created by an impartial researcher. Abbreviations: 6MWT, 6 min walking test; CPET, cardiopulmonary exercise test; ISWT, incremental shuttle walking test.

**Figure 2 jpm-12-00901-f002:**
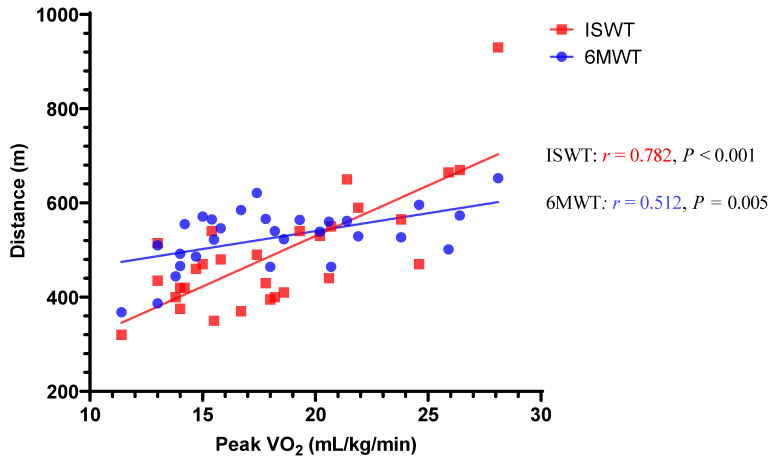
The correlation analysis of ISWT and 6MWT in 29 patients in both hospitals. The validity of ISWT and 6MWT was analyzed using Pearson’s correlation coefficient (*r*). Both walking tests presented a significant correlation with peak VO_2_ of the cardiopulmonary exercise test. However, ISWT presented a higher correlation than 6MWT. Abbreviations: 6MWT, 6 min walking test; ISWT, incremental shuttle walking test; peak VO_2_, peak oxygen consumption.

**Figure 3 jpm-12-00901-f003:**
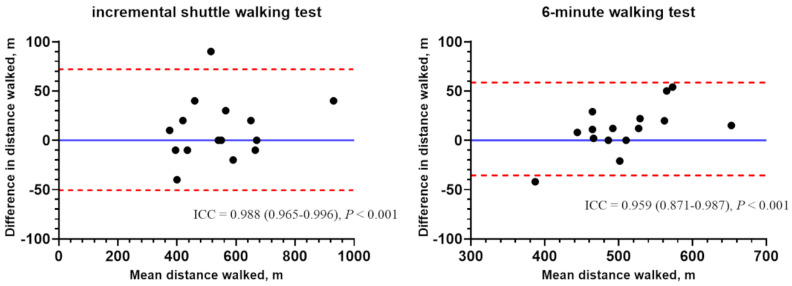
Bland–Altman plots representing reliability between the first and second tests of incremental shuttle walking test and 6 min walking test in Ulsan University Hospital. ICC was used to evaluate reliability. Abbreviation: ICC, intraclass correlation coefficients.

**Figure 4 jpm-12-00901-f004:**
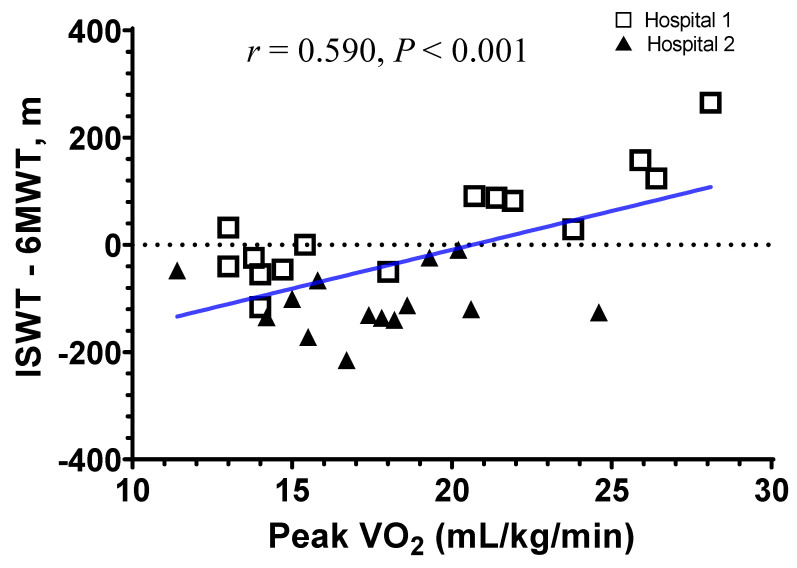
The correlation between peak VO_2_ and the difference of walking distance between ISWT and 6MWT. The correlation coefficient (*r*) was analyzed using Pearson’s correlation. Patients with higher peak VO_2_ walked a longer distance in ISWT than 6MWT. Abbreviations: 6MWT, 6 min walking test; ISWT, incremental shuttle walking test; peak VO_2_, peak oxygen consumption.

**Table 1 jpm-12-00901-t001:** Baseline characteristics of study participants.

	Total (*n* = 29)	Hospital 1 (UUH) (*n* = 15)	Hospital 2 (AMC) (*n* = 14)	*p* ^†^
Age (years) *	67.0 (61.0–72.0)	65.0 (61.0–71.0)	67.5 (64.0–73.5)	0.619
Male gender	28 (96.6%)	15 (100.0%)	13 (92.9%)	0.483
Body mass index (kg/m^2^) *	23.7 (22.3–26.0)	23.1 (22.1–26.7)	24.3 (23.2–25.5)	0.680
Smoking history				0.483
Nonsmoker	1 (3.4%)	0 (0.0%)	1 (7.1%)	
Current or ex-smoker	28 (96.6%)	15 (100.0%)	13 (92.9%)	
Smoking amount (pack-years)	35.0 (26.0–49.0)	39.0 (30.0–50.0)	34.0 (14.5–49.0)	0.400
Comorbidity				
DM	3 (10.3%)	3 (20.0%)	0 (0.0%)	0.224
HTN	7 (24.1%)	3 (20.0%)	4 (28.6%)	0.682
Pulmonary HTN	1 (3.4%)	1 (6.7%)	0 (0.0%)	>0.999
CAD	1 (3.4%)	0 (0.0%)	1 (7.1%)	0.483
Arrhythmia	1 (3.4%)	0 (0.0%)	1 (7.1%)	0.483
Lung cancer	1 (3.4%)	0 (0.0%)	1 (7.1%)	0.483
Malignancy except lung	4 (13.8%)	2 (13.3%)	2 (14.3%)	>0.999
Any exacerbation in the past year	7 (24.1%)	6 (40.0%)	1 (7.1%)	0.080
Inhalers				0.041
No use	1 (3.4%)	0 (0.0%)	1 (7.1%)	
LAMA	1 (3.4%)	0 (0.0%)	1 (7.1%)	
LAMA + LABA	17 (58.6%)	6 (40.0%)	11 (78.5%)	
LABA + ICS	2 (6.9%)	2 (13.3%)	0 (0.0%)	
LAMA + LABA + ICS	8 (27.6%)	7 (46.7%)	1 (7.1%)	
mMRC	1.0 (1.0–2.0)	1.0 (1.0–1.0)	1.5 (1.0–2.0)	0.046
CAT *	7.0 (4.0–12.0)	8.0 (5.0–12.0)	6.5 (3.8–17.5)	0.637
Pulmonary function test				
FEV1 (L) *	1.9 (1.4–2.3)	2.1 (1.3–2.3)	1.8 (1.4–2.2)	0.409
FEV1, % predicted	66.5 (54.5–82.5)	66.5 (53.9–82.7)	67.9 (55.9–83.5)	0.914
FVC (L) *	3.8 (3.3–4.3)	3.8 (3.5–4.3)	3.6 (3.1–4.3)	0.362
FVC, % predicted	108.8 (94.7–117.4)	108.8 (93.8–115.4)	108.9 (95.0–119.5)	0.914
FEV1/FVC	47.0 (38.5–59.5)	47.0 (35.0–62.0)	51.0 (42.3–59.0)	0.780
CPET				
Peak VO_2_ (mL/kg/min)	17.8 (14.5–21.1)	18.0 (14.0–23.8)	17.6 (15.4–19.5)	0.780
MVV *	73.0 (47.5–97.5)	87.0 (46.0–99.0)	72.0 (54.0–95.5)	0.525
Breathing reserve *	19.0 (3.0–33.5)	14.0 (2.0–21.0)	29.2 (14.8–38.3)	0.105
Peak RQ	1.3 (1.2–1.4)	1.2 (1.1–1.4)	1.4 (1.3–1.5)	0.188
O_2_ pulse	10.0 (8.5–12.6)	9.6 (7.4–13.1)	10.3 (9.0–12.5)	0.161
Distance of ISWT (m) *	483.5 ± 124.1	538.7 ± 143.7	424.3 ± 60.3	0.011
Distance of 6MWT (m) *	525.3 ± 62.5	502.5 ± 57.5	549.8 ± 60.2	0.040

All values are presented as *n* (%) or median (interquartile range) unless otherwise stated. ^†^ *p* value was analyzed using the *χ*^2^ or Fisher’s exact test for categorical variables and the Mann–Whitney *U* test for continuous variables to compare difference between Hospital 1 and Hospital 2. We presented the distance of ISWT and 6MWT as the mean and standard deviation. * *p* values were analyzed using Student’s *t*-test, because these variables presented a normal distribution based on the Kolmogorov–Smirnov test. Abbreviations: 6MWT, 6 min walking test; AE, acute exacerbation; CAT, chronic obstructive pulmonary disease assessment test; CPET, cardiopulmonary exercise test; DM, diabetes mellitus; HTN, hypertension; FEV_1_, forced expiratory volume in one second; FVC, forced vital capacity; ICS, inhaled corticosteroid; ISWT, incremental shuttle walking test; LABA, long-acting β_2_ agonist; LAMA, long-acting muscarinic antagonist; mMRC, Modified Medical Research Council; MVV, maximum voluntary ventilation; peak VO_2_, peak oxygen consumption; RQ, respiratory quotient.

**Table 2 jpm-12-00901-t002:** Comparison of the correlation between peak oxygen uptake and distance of ISWT and 6MWT.

	Total (*n* = 29)		Hospital 1 (*n* = 15)		Hospital 2 (*n* = 14)		*p **
	*r*	*p*	*r*	*p*	*r*	*p*	
Peak VO_2_~ISWT	0.782	<0.001	0.868	<0.001	0.540	0.046	0.084
Peak VO_2_~6MWT	0.512	0.005	0.685	0.005	0.572	0.033	0.653
*p* ^†^	0.043		0.029		0.883		

Correlation coefficient (*r*) was analyzed using Pearson’s correlation. * *p* value was analyzed using the Fisher’s *z* transformation; ^†^ *p* value was analyzed using Dunn and Clark’s *z* test. Abbreviations: ISWT, incremental shuttle walking test; 6MWT, 6 min walking test; peak VO_2_, peak oxygen consumption.

## Data Availability

The datasets used and/or analyzed during the current study are available from the corresponding author upon reasonable request.

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
