# Peer review of "Correlation Comparison and Personalized Utility of Field Walking Tests in Assessing the Exercise Capacity of Patients with Chronic Obstructive Pulmonary Disease: A Randomized Controlled Trial"

_jpm, 2022, doi:10.3390/jpm12060901_

Round 1
Reviewer 1 Report
In this study, Ko et al. investigated the validity and reliability of incremental shuttle walking test comparing with CPET in COPD patients. I have the following comments:
- The majority of included patients were male, so the conclusion cannot be drawn for female COPD patients. This has to be declared.
- Table 1: Does the pvalue represent the difference between the two hospitals or overall?
- Table 1: breathing reserve was obviously higher in hospital two.
- In hospital one, ISWT was greater than 6MWT while it was contrary in the hospital two. Furthermore, there were quite comparable VO2 peaks with different distances in ISWT and 6MWT in both hospitals. This can also been seen in Table 2, where r=0.54 and r=0.87 are quite different among the hospitals. Although the authors try to discuss this issue, I wonder if the finding could be due to a methodological issue.
- Although the ICC is statistically high, there are various patients with differences in distance walked above the MID. This should be discussed.
- Conclusion: It does not surprise that two different tests yield different walking distances. The further research question should be if the ISWT is a similar or better predictor for relevant outcomes (e.g. mortality) than the 6MWT.
- Table S2. Median Peak SBP in hospital two was 215.5. This is quite high and would mostly lead to test interruption.
- Table S2. There is an obvious difference in post borg scale between hospital 1 and 2 according to 6MWT that is in line with the walking distances of the respective tests. Could this be an explanation for the lower 6MWD in hospital 1? It is unusal to have Borg 1 after a submaximal exercise capacity test.
Reviewer 2 Report
1. In the sample size calculation, why did the authors choose to use 0.6 as the hypothesized correlation, rather than another relatively low correlation coefficient? Any evidence-based or prior studies made a decision in 0.6?
2. In Table 1, the authors used median (IQR) and Mann-Whitney U tests to present baseline characteristics of study participants, with non-normal distributions by default for continuous data. The authors performed Pearson's correlation for all correlation analyses, but Pearson's correlation applies to normally distributed continuous data. Therefore, the authors may change the nonparametric correlation analysis or log-transform continuous data.
3. Univariate analysis alone cannot draw conclusions about the association between ISWT, 6MWT, and CPET and should be validated in multivariate analysis.
Round 2
Reviewer 1 Report
I have no further comments. The previous comments were addressed in the revised version of the manuscript.
Author Response
We would like to thank all the reviewers and editors who reviewed our manuscript and provided valuable feedback that undoubtedly helped us improve the manuscript. We made every effort to respond to all questions and comments; the following are point-by-point responses to each of them. Please let us know if any of our responses are unsatisfactory, and we will work to improve them.
I have no further comments. The previous comments were addressed in the revised version of the manuscript.:
We would like to sincerely thank you for your previous review and comments.
Reviewer 2 Report
minor comment:
ISWT and 6MWT should be presented as mean +- standard deviation and tested by the student t test in Table 1.
